# Household Flood Severity and Migration Extent in Central Java: Analysis of the Indonesian Family Life Survey

**DOI:** 10.3390/ijerph20095706

**Published:** 2023-05-02

**Authors:** Breanne K. Langlois, Leah Beaulac, Katherine Berry, Oyedolapo Anyanwu, Ryan B. Simpson, Aris Ismanto, Magaly Koch, Erin Coughlan de Perez, Timothy Griffin, Elena N. Naumova

**Affiliations:** 1Friedman School of Nutrition Science and Policy, Tufts University, Boston, MA 02111, USA; 2Department of Oceanography, Faculty of Fisheries and Marine Science, Universitas Diponegoro, Semarang 50275, Indonesia; 3Center for Remote Sensing, Department of Earth & Environment, Boston University, Boston, MA 02215, USA; 4Red Cross Red Crescent Climate Centre, 2502 KC The Hague, The Netherlands

**Keywords:** flooding, disasters, climate change, migration, adaptation, vulnerability

## Abstract

Central Java, Indonesia, is prone to river and coastal flooding due to climate changes and geological factors. Migration is one possible adaptation to flooding, but research is limited due to lack of longitudinal spatially granular datasets on migration and metrics to identify flood-affected households. The available literature indicates social and economic barriers may limit mobility from flood prone areas. The Indonesian Family Life Survey (IFLS) provides self-reported data on household experiences with natural disasters among 1501 Central Java households followed over two waves (2007 and 2014). We examined how the severity of flooding, defined by household-level impacts captured by the IFLS (death, injury, financial loss, or relocation of a household member), influenced the extent of household movement in Central Java using a generalized ordered logit/partial proportional odds model. Households severely impacted by floods had 75% lower odds of moving farther away compared to those that did not experience floods. The most severely impacted households may be staying within flood-affected areas in Central Java. Public health, nutrition, and economic surveys should include modules focused on household experiences, impacts, and adaptations to facilitate the study of how climate changes are impacting these outcomes.

## 1. Introduction

Climate-related migration is directly linked to public health, as migrants and those left behind face food insecurity, malnutrition, and reduced access to services among other health threats [1,2,3]. While migration out of degraded areas is one example of adaption to climate change, the main push factors are often deteriorating livelihoods and economic situations [4]. There is global political interest in addressing these interconnected issues, with calls to prioritize migrant health interventions and include migration in national climate change plans during the 26th Conference of the Parties (COP26) for the United Nations Framework Convention on Climate Change [3].

While there is global attention to the issue of climate migration, especially in island nations experiencing frequent flooding from sea level rise and extreme weather, there are significant challenges to research in this area [5,6]. According to the most recent report from the International Organization for Migration (IOM), the various forms of migration across space and time (e.g., within a village or internationally, temporary or permanent), in addition to multiple push and pull factors, and the lack of suitable datasets make climate migration a difficult phenomenon to define [5]. While the “Migration, Climate Change, and the Environment” bibliographic database (CliMig) collates case studies related to climate migration, there is a lack of longitudinal and spatially granular data on migration to evaluate the multiple drivers and slow-onset environmental processes [7]. Some surveys, including the Indonesian Family Life Survey (IFLS), ask respondents directly about displacement due to disasters, but such information only captures the immediate post-disaster response. Further, while flooding affects more people worldwide than any other climate-related disaster, drought and desertification remain the most studied hazards in the CliMig database [5].

In areas such as Central Java, flooding caused by both coastal inundation and river overflow from heavy precipitation is a regular occurrence [8,9,10]. Many studies on the impact of floods in the Indonesian context examine either coastal or river flooding or focus more broadly on sea level rise [11,12,13,14,15,16]. More research is needed on how all forms of flooding impact long term migration. The Internal Displacement Monitoring Centre (IDMC) estimated 155,000 internally displaced persons from disasters in Indonesia in 2021, but this figure does not distinguish temporary versus permanent movement or the main cause [17].

The literature indicates there may be substantial barriers to migration among poor, vulnerable households where climate hazards may limit capacity to move, challenging common assumptions about migration [18,19,20,21,22,23,24]. In one of the first robust analyses using longitudinal data to examine flood related migration in rural Bangladesh, researchers found flooding to have a minimal impact, while non-flood related crop failure had a large effect [19,20]. Studies from Indonesia show a preference to stay in affected areas and adapt [15]. Migration out of flood-affected areas is often a last resort due to social factors and the economic burden of migration, especially if the flooding is gradual or not severe [4,15]. Local adaptations to “the Rob”—a term used to describe the regular occurrence of seawater overflow during tides—include raising floor levels using concrete or stilts, using floating structures, or creating dams to block water from entering households [11,15]. This research points to the need to examine the household-level impacts and how vulnerability, economic barriers, and adaptive capacity may shape migration; however, identifying flood-affected households is challenging, and there is no metric of flood severity that incorporates the household-level impacts.

Self-reported data on experiences with natural disasters from the IFLS allows for the identification and study of flood-affected households over a 12-year span captured through two of the five cross-sectional survey waves. As a nationally representative, longitudinal economic and health survey that tracked Indonesian households from 1993 to 2014, the IFLS is robust in reflecting the overall population with its multi-stage sampling scheme, large sample size (15,921 interviewed households including 1949 in Central Java), and high tracking rate (94%) [25]. Other strengths of the IFLS include its multipurpose nature, extended time period (21 years), and availability of both current and retrospective data.

We examined how the severity of flooding, defined by household-level impacts of floods, influenced the extent of household movement in Central Java in the 1501 IFLS households that were tracked over two waves in 2007 and 2014. Specifically, we: (1) describe flooding and other type of disasters experienced by households and their reported direct impacts, and (2) evaluate how the severity of flooding, agricultural livelihood, and socio-economic characteristics influenced the extent moved. We discuss usability and data quality aspects of the IFLS for this purpose. This paper provides insight into how flooding disasters are impacting households in a largely agricultural area, Central Java, and may help with disaster planning and mitigation efforts.

## 2. Materials and Methods

### 2.1. Design of the Indonesian Family Life Survey

The RAND Corporation and Survey Meter conducted the IFLS. Ethical approval was obtained by the RAND Institutional Review Board in the United States and the University of Gadjah Mada Institutional Review Board in Indonesia. The IFLS is designed to study a wide range of behaviors and outcomes at individual and household levels including economic, well-being, consumption, income and wealth, education, and health indicators.

The IFLS is a longitudinal, population-based survey that was conducted over five cross-sectional waves fielded in 1993, 1997, 2000, 2007, and 2014. Due to the longitudinal design, sampling for the IFLS was based on wave one, with each subsequent wave attempting to contact and reinterview this sample. The original wave one survey sample from 1993 used a two-stage stratification process based on province and urban/rural location [26]. Households were then randomly selected from within each stratum. For cost reasons, the IFLS sampled from only half of the provinces. A total of 7224 households from 13 of 27 provinces participated in 1993, representing ~83% of the country’s total population at the time. In addition to these “origin” households, the IFLS tracked newly formed households that contained a target member of an original household. The IFLS collected detailed demographic, economic, livelihood, and health information across all five waves from 6044 households (84% of the 1993 sample), including 828 from Central Java (94% of the 1993 Central Java sample). The survey incorporated a natural disasters module in waves four (2007) and five (2014). This module captured self-reported experiences with natural disasters in the prior five years, including the severity and impacts. More detailed information about the survey including data collection and verification process, instruments, and data cleaning can be found in the user guides [26].

A total of 1501 Central Java households were tracked across waves four and five (13% of all interviewed households followed over both waves, 93% tracking rate of wave four households in wave five). Of these, we analyzed 1472 (98%) that had complete data. We estimate our detectable effect size is 10 percentage points based on our sample size, an 8:1 group ratio, and 20% migrating among those not affected by flood.

### 2.2. Outcomes and Predictors

The main outcome was the extent the household had moved since the time of the last survey, between 2000 and 2007 for wave four and between 2007 and 2014 for wave five. This was a 6-level ordinal variable defined as: (0) the household did not move; or the household moved (1) within the same village, (2) within the same district, (3) within the same regency, (4) within the same province, or (5) to another province. We obtained this variable from the household tracking files provided with IFLS data. Field interviewers tracked households by returning to the address where the household was in the previous survey and, if the entire household was missing, locating all target members. While the IFLS included a module with direct questioning about the impacts of flooding and other disasters, including time spent without housing, it only captured the immediate response of some household members. Our interest was in the longer-term impact. Therefore, we used whole-household moves as our measure of migration.

All independent variables were self-reported from the household survey. Severity of flooding experienced by households was the main independent variable, defined as: (0) the household did not experience a flood disaster; (1) experienced a flood disaster, but the impacts were not severe; or (2) experienced a flood disaster that had severe impacts. We identified households that experienced floods by the survey questions of whether there were any natural disasters in the area where the households resided in the prior 5 years and by the reported types of disasters. The IFLS Natural Disaster module captured both natural disasters and events that cause economic disruption—such as floods, earthquakes, tsunamis, mudslides, civil strife, death of household head or member, serious illness, job loss, etc.—collected from a check-all-that-apply list with 18 possible response options (See Appendix A for a complete listing). “Severity” was predefined by a question within the IFLS Natural Disaster module of whether any disasters were “severe enough to cause death or major injuries of a household member, cause direct financial loss to the household, or cause household members to relocate”. The second independent variable of interest was whether the household was agricultural. To be as broad as possible in our definition of an agricultural household, we constructed an indicator of whether the household had any members engaged in a farm business or activity. See Appendix A for the metrics and corresponding survey questions.

The rural poor may be more vulnerable to negative impacts of flooding according to prior research [27]. We evaluated the relationship to migration by including cofactors collected in the Household Characteristics module in the model. These were urban or rural areas, household size, main water source, type of toilet, sewage, and garbage disposal systems as measures of socio-economic status. Each of these factors may also be associated with a household’s experience with flooding, as they may be determinants of living in areas more prone to flooding.

Cross-tabulations of each categorical independent variable stratified by the ordinal outcome variable showed exceedingly low cell counts (<15), and the model with the six-level ordinal outcome did not converge; therefore, we collapsed it into a five-level category for modelling defined as: (0) the household did not move, or the household moved (1) within the same village, (2) within the same district, (3) within the same regency, (4) outside of the regency or to another province. Similarly, due to low frequencies in the response categories, we collapsed other socio-economic characteristics into fewer categories or into binary variables for modelling as follows: type of toilet to a binary variable indicating owning a septic system, type of sewage draining system to a binary variable indicating flowing drainage ditch, type of garbage disposal system to a three-level categorical variable (collection by sanitation service/burned/or other), and type of drinking water to a four-level categorical variable (pipe/well with pump/well water/or other).

### 2.3. Data Analysis

All data management and analysis were done using Stata 17 (Stata Statistical Software: College Station, TX, USA: Stata Corp LP). We calculated percentages and 95% confidence intervals of the types of disasters, including floods and related impacts, socio-economic and demographic characteristics for the wave four and wave five cross-sections. To evaluate whether occurrence and severity of flooding influenced migration, we fit a generalized ordered logit and partial proportional odds model as follows [28]:(1)$$PY_{i}>j=\frac{\exp\left(\beta_{0_{j}}+{\beta_{1}X}_{i,u}+\beta_{2_{j}}Z_{i,v}\right)}{1+[\exp\left(\beta_{0_{j}}+{\beta_{1}X}_{i,u}+{\beta_{2_{j}}Z}_{i,v}\right)]}$$
where *i* represents household, and *j* represents the cumulative logit models for the M − 1 levels of the ordinal outcome *Y* (with M representing the number of levels).

The model produced four sets of coefficients for the four cumulative logit models based on the five-level ordinal outcome, specifically for comparing the odds of (1) not moved versus moved; (2) not moved or moved within the same village versus moved out of the village; (3) not moved or moved within the same district versus moved out of the district; and (4) not moved or moved within the same regency versus moved out of the regency. This is a more parsimonious alternative to the ordered logit model, which relies on the assumption of proportional odds (i.e., the relationship between the independent and dependent variables are the same for each of the cumulative logistic regressions). The partial proportional odds model allows for beta coefficients that are both constrained and unconstrained to the proportional odds assumption. For independent variables that meet the assumption, one coefficient is given. For variables that do not meet the assumption, four sets of coefficients are given. The independent variables, such as household experience with flood, indicator of agricultural livelihood, socio-economic characteristics, indicator of whether the household was in an urban or rural area, household size, and survey wave are denoted by *u* and *v*. In the above equation, ${\beta_{1}X}_{i,u}$ represents the coefficients for the independent variables, *u*, that are the same across the values of *j*, and $\beta_{2_{j}}Z_{i,v}$ represents the coefficients for the independent variables, *v*, that are not the same across the values of *j*. We present odds ratios and interpret them based on direction, with higher odds indicating leaving the locality and lower odds indicating staying within the locality. We determine statistical significance at the α = 0.05 level.

We fit univariate models for each independent variable before fitting the full, adjusted model. We used complete case analysis (less than 1% missing for each variable included in the model). We examined the collinearity of independent variables using variance inflation factor (with values of 5 or higher considered problematic), as well as assessing changes in model output between univariate and adjusted models. There was no evidence of multicollinearity, with all VIF <2. Other disaster variables, including household experience with multiple disasters, were not included in the models due to very little variation (<3%).

## 3. Results

Descriptive statistics are presented in Table 1. The population can be characterized as predominantly rural (>55%), with most households having running water and a septic system. About 40% were agricultural households, with at least one household member working as a farmer.

### 3.1. Reported Types of Disasters and Their Impacts

Over the time span, 31% (466) of Central Java households reported being affected by a select disaster (Appendix A) in the area where they live, with 3% (52) experiencing multiple disasters. Floods were the most common disaster, involving 13% (190) of Central Java households over the period, followed by earthquakes at 6% (92), volcanoes at 4% (58), windstorms at 3% (49), drought at 2% (33), and fire at 1% (17) (Table 1). A total of 7% (104) of Central Java households (22% of disaster-affected households) reported being affected by a disaster severe enough to result in death, significant injuries, financial loss, or relocation. These severely impacted households were mainly affected by flood at 55% (58), followed by an earthquake at 21% (22), windstorms at 13% (13), drought at 10% (10), and volcanoes at 9% (9).

In addition to the types of disasters above, respondents reported the direct impacts of those disasters from a set of questions in the Natural Disaster module (Appendix A). Of the 58 households that were severely impacted by floods, 17% (10) had multiple disasters at the same time, mostly failed harvests (5). This subpopulation did not indicate damage or destruction of homes due to flooding, but temporary displacement was standard with 60% (35) reporting some time without housing or in temporary housing. All who spent time without housing indicated they did return or planned to return to their home. Most households (55%) received some government or NGO assistance. Most assistance was from the regional government (56% of households that received assistance). No deaths, injuries, or illnesses due to the flood were reported by these households.

### 3.2. Household Migration Response

Most households, 72% (1081), did not move during the time span, while 28% (420) moved at least once; 21% (309) moved once and 7% (111) moved in both waves.

Table 2 displays results of the generalized ordered logit/partial proportional odds model. The model included 1472 households with complete data. The proportional odds assumption held for flood severity, flowing drainage system, and owning a septic system, but was not imposed for agricultural livelihood, type of garbage disposal system, water source, household size, rural area, and survey wave. Compared to households not exposed to floods, those that were exposed to floods with severe impacts had 75% lower odds of moving farther away (OR 0.25, 95% CI 0.09, 0.70) and agricultural households had 42% lower odds (OR 0.58, 95% CI 0.44, 0.77). Drinking water source, household size, and rural area were associated with the extent moved. Compared to households with piped water, there were higher odds of moving farther away among those with pumped well water (OR 1.36, 95% CI 1.01, 1.83) and other water sources including spring, rain, or river (OR 1.93, 95% CI 1.42, 2.61). Rural households had 62% lower odds of moving farther away (OR 0.38, 95% CI 0.29, 0.51). An increase in household size had 7% lower odds of moving farther away (OR 0.93, 95% CI 0.87, 0.99). Compared to the period from 2002 to 2007, the odds of moving farther away from 2009 to 2015 decreased by 56% (OR 0.44, 95% CI 0.35, 0.55).

## 4. Discussion

### 4.1. Migration Extent

This study assessed how flooding and the severity of impacts effected the extent of household movement in Central Java, Indonesia using data from a household survey. We distinguished between local and long-distance moves by using the household migration extent as our outcome. We quantified flood severity using information about the reported impacts to households. We found households that are agricultural, rural, and those most severely affected by floods had lower odds of moving farther away. In other words, they tended to stay locally.

In a similar study in 2014 by Bohra-Mishra et al., which analyzed farmers using four waves of IFLS data (1993–2007), the findings indicated that precipitation, which can be a precursor to flooding events, had a smaller impact on migration than temperature change [29]. Results from the IFLS are consistent with other Indonesia studies indicating a tendency to prioritize adaptation strategies other than migration when faced with flooding disasters [15,29,30]. In the IFLS, almost all who reported spending some time without housing due to flood indicated they returned. In a 2011 study in Semarang, Indonesia, by Harwitasari et al., the findings indicated a deep connection to dwellings and noted that abandoning their homes was not realistic [30]. The study found that community members, especially homeowners, were willing to pay and coordinate together on adaptation practices to remain on their land. In a 2018 study by Buchori et al. in Semarang, Indonesia, 81% of respondents indicated a preference to stay rather than migrate [15]. The most reported reason for remaining was an emotional attachment to their homes (41%), followed by proximity to work (34%) [15]. For households that did decide to migrate, a significant proportion (44% of the 19% that reported migrating) chose local hillier areas within the Semarang area [15]. Knowledge and policy gaps related to migration in the context of slow-onset impacts of climate change remain [4,5].

Our finding that severely affected households tended not to move as far away may signify a turning point where adaptation and adaptive capacity—including a household’s ability to relocate out of degraded areas—may be diminished. These findings are consistent with other studies and with migration theory that the losses from flooding disasters may make longer distance moves more difficult [15,18,19]. This hypothesis is reinforced by the most recent IPCC report which detailed projected limits to adaptation in coastal and agricultural communities as global warming continues [4]. The losses caused by floods may create “trapped” populations, as households that are most affected become the ones who lack the capital needed to start a new life somewhere else. This phenomenon has been documented in other flood-affected settings [6]. In our study, findings indicated that the most vulnerable were larger, rural, agricultural households.

Floods were the most reported disaster to be severe enough to result in death, major injuries, financial loss, or relocation. Flooding in the Indonesia context can carry a multitude of definitions, all resulting in potential implications for agricultural households. For example, “the Rob”, which refers to the combination of temporary flooding and mostly permanent inundation, is common in the northern coastal region of Java [15]. Additionally, Indonesia is vulnerable to various compounding climate change factors such as sea level rise and variability in intensity and duration of monsoon rains [31]. In a 2011 study by Syaukat analyzing the impact of climate change on food production in Indonesia, the analysis demonstrated significant changes in wet and dry seasons when comparing periods between 1961 and 1990, and 1991 and 2003, 10–20 days later and 10–60 days earlier, respectively [32]. This study found climate change related factors could lead to an estimated deficit of 90 million tons of husked rice by 2050. Considering the variety of climatic factors related to flooding that may impact agriculture in Indonesia, future studies should seek to better understand and differentiate between type and severity of floods. While most studies are specific to either riverine or coastal flooding, the measure used in this study encompassed all types of floods and parsed out severely affected households.

### 4.2. Usability Aspects of the IFLS

The IFLS was designed for studying various aspects of social and economic well-being of the Indonesian population and is a feasible data source for studying how environmental factors impact households. The self-reported information on natural disasters over two survey waves in 2007 and 2014 with respondent recall spanning 10 years and with direct questioning on the impacts allows users to understand the immediate damages. Beyond the immediate damages, users can use the IFLS to explore how disasters affected other aspects of life through the breadth of information related to livelihoods, expenditures, migration, and health at individual, household, and community levels. The IFLS data are provided in multiple formats in several modules, requiring user processing to reshape and merge selected files. The IFLS maintains consistency of variables throughout survey waves, facilitating cross-sectional or longitudinal data analyses. The IFLS has spatial aggregation at the district-level.

The IFLS allows for identification of flood-affected households directly through self-report. The IFLS most likely captures sudden-onset events because it measures a recall of disasters during the previous five years and does not provide a good measure of recurrent, or regular, flooding and the slow-onset changes. Another similar study also used direct survey responses to measure floods, however, as discussed in Oskorouchi et al., this measure may be biased, as those who were more affected may have been more likely to report a flood [33]. The self-reporting of floods also has limitations in terms of precision. Other studies have used a range of approaches to classify flooding experienced by households, including linking rainfall, disaster, and other spatial data, or using satellite precipitation-driven modeling [15,33,34,35,36]. In a previous study of the IFLS, the authors linked IFLS data with the DesInventar database, which provides information on disasters from official sources, academic records, newspapers and institutions [29]. While these data linkages are possible with the IFLS, there is risk of misclassification of flood-affected households, as the smallest administrative boundary is at the district level.

While the IFLS captures all types of flooding, including river and coastal flooding, it did not delineate between types of floods or main cause. The survey tool asks whether households experienced a disaster (flood or other type of disaster), how many times, and when was the most severe event (see Appendix A). The information about the frequency of events could have provided another measure of flood but with very low frequencies when subset to Central Java due to a logical skip pattern in the survey. Findings from prior studies indicate that flood severity does impact the types of adaptation strategies chosen, such as migration or household modifications [15]. We classified flood severity based on a survey question of whether the disaster was severe enough to result in death, major injuries, financial loss, or relocation, but these impacts could not be disaggregated; however, this information did allow for identification of the most severely impacted households from which to derive a flood severity metric.

The IFLS had an excellent tracking record of households across survey waves, allowing for an accurate measure of whole household movement. Findings from our study differ from the Internal Displacement Monitoring Centre (IDMC) database, which reports consistently higher displacement figures as a result of sudden-onset disasters, primarily flooding, throughout the IFLS survey period [17,37]. This is likely because we examined whole-household migration rather than temporary displacement. While the IFLS asked whether any household members spent time without housing due to a disaster, it did not ask directly if households relocated as a result. Further, the direct questioning captures the immediate rather than long-term impacts. Reasons why households stayed or moved were not captured by the IFLS but would enhance understanding of vulnerability and immobility in the context of flooding disasters.

Due to the cross-sectional nature of the data, it could not be determined whether floods preceded household move, leading to potential antecedent-consequent bias. This may have the effect of biasing toward the null, meaning there may be a larger effect than what was detected in this study. The IFLS contained an individual-level module on migration, which was not examined in our study, that is suited towards studying the social and economic drivers. This data could be explored more to understand the various drivers, including slow-onset environmental changes, which a previous paper has examined through data linkages [29].

## 5. Conclusions

Flood-affected households in Central Java that suffered losses tended to stay locally compared to those that were unaffected by floods, while flood-affected households that did not suffer losses showed no difference. We found differences in the extent moved in households that were agricultural and rural, which also tended to stay locally. These findings provide evidence that more vulnerable households may be staying within flood-affected areas in Central Java. Immobility in the context of climate hazards should be further researched and better understood. Future surveys should incorporate experiences with small, gradual shifts in the environment and compounding effects of multiple disasters. Disaggregation in data, and evaluating non-linearity in relationships, can allow for deeper understanding of how environmental disasters are impacting lives.

## Figures and Tables

**Table 1 ijerph-20-05706-t001:** Description of Central Java Households.

Household Characteristics	Wave 4, 2007 (n = 1458)		Wave 5, 2014 (n = 1472)	
	%	(n)	%	(n)
Rural	58	(846)	55.2	(813)
Agricultural	39.9	(582)	37	(544)
Main source of drinking/cooking water				
Pipe water	22.4	(327)	27.9	(411)
Well/pump (electric, hand)	27.4	(400)	26.6	(392)
Well water	22.9	(334)	9.9	(145)
Spring water	15.1	(220)	12.2	(179)
Rainwater	0.6	(9)	0.1	(2)
River/Creek water	0.3	(5)	0.5	(8)
Water collection basin	2.2	(32)	0.4	(6)
Aqua/Mineral water	6.9	(101)	15.9	(234)
Other	0.2	(3)	0.9	(13)
Type of toilet/Where household members defecate				
Own toilet with septic	61.4	(895)	69.4	(1021)
Own toilet without septic	8.8	(129)	9	(133)
Shared toilet	6.7	(98)	4.2	(62)
Public toilet	3.2	(47)	3.1	(46)
Creek/river/ditch (without toilet)	11.5	(168)	4.3	(64)
Yard/field (without toilet)	0.6	(9)	0.5	(8)
Pond/fishpond	4.9	(71)	2.6	(39)
Sea/lake	0.8	(11)	0.3	(5)
Other	0.1	(2)	0.9	(13)
Sewage disposal system				
Drainage ditch (flowing)	61.2	(892)	54.4	(801)
Drainage ditch (stagnant)	2.1	(30)	2.6	(39)
Permanent pit	10.2	(149)	12.8	(189)
Disposed into river	7.1	(103)	7.5	(110)
Disposed in yard/garden	7.3	(106)	6.6	(97)
Pond/Fishpond/Lake/Pool	6.8	(99)	5.6	(83)
Hole (without permanent lining)	3.2	(46)	3.8	(56)
Paddy or other field	0.3	(5)	0.5	(8)
Other	0.1	(1)	0.5	(8)
Type of garbage disposal system				
Trash can, collected by sanitation service	23	(335)	24.8	(365)
Burned	37	(540)	44.5	(655)
River/Creek/Sewer	7.7	(112)	8.1	(119)
Disposed in yard and let decompose	14.9	(217)	11.5	(169)
Pit	10.8	(157)	2.8	(41)
Forest, mountain	0.5	(7)	0.1	(1)
Sea, lake, beach	1.2	(18)	1.2	(18)
Paddy or other field	2.4	(35)	0.5	(8)
Other	0.7	(10)	1	(15)
Experience with disasters				
Experienced any disaster in area where household lives in past 5 years	14.4	(210)	19.5	(287)
Flood	5.2	(76)	8.4	(123)
Landslide	2.1	(31)	2.9	(43)
Volcanic	0	(0)	3.8	(56)
Earthquake	4.3	(63)	1.6	(24)
Windstorm	0.8	(11)	2.7	(40)
Fire	0.1	(2)	0.8	(12)
Drought	0	(0)	2.2	(33)
Experienced multiple disasters	0.9	(13)	2.6	(38)
Flood severity				
Did not experience a flood disaster	92.9	(1355)	86.1	(1267)
Experienced a flood disaster with no severe impacts	2.9	(43)	6.9	(101)
Experienced a flood disaster with severe impacts	2.3	(33)	1.5	(22)
Migration outcome				
Extent moved since previous survey				
Household did not move	79.7	(1162)	88.5	(1302)
Moved within same village	1.6	(23)	2.2	(33)
Moved within same district	3.4	(49)	2.1	(31)
Moved within same regency	5.8	(84)	1.8	(26)
Moved within same province	5.1	(75)	2.3	(34)
Moved to another province	4.3	(62)	3	(44)

**Table 2 ijerph-20-05706-t002:** Generalized ordered logit/partial proportional odds model assessing the extent moved among Central Java households ^1,2^.

	Odds Ratio	SE	*p*-Value	Lower Bound	Upper Bound
* Main variables of interest*					
Flood severity (no experience with flood is ref.)					
Experienced a flood with no severe impacts	0.93	0.21	0.745	0.60	1.44
Experienced a flood with severe impacts	0.25	0.13	0.008	0.09	0.70
Agricultural household	0.58	0.08	0.000	0.44	0.77
* Other household socioeconomic factors*					
Owns toilet with septic	1.23	0.16	0.096	0.96	1.58
Sewage drains from flowing drainage ditch	0.88	0.10	0.260	0.70	1.10
Garbage disposal (collection by sanitation service is ref.)					
Burned	1.09	0.17	0.581	0.81	1.47
Other (river, yard, pit, etc.)	0.91	0.15	0.558	0.66	1.25
Main source of drinking/cooking water (pipe is ref.)					
Well, pump (electric, hand)	1.36	0.21	0.043	1.01	1.83
Well water	1.32	0.22	0.104	0.95	1.84
Other (spring, rain, river, etc.)	1.93	0.30	0.000	1.42	2.61
Household size	0.94	0.03	0.077	0.89	1.01
Rural	0.38	0.06	0.000	0.29	0.51
Survey wave (wave 5 vs. 4)	0.49	0.05	0.000	0.40	0.60
Constant	0.42	0.10	0.000	0.27	0.66
Variables not constrained to the proportional odds assumption					
Model 2 ^2^					
Agricultural household	0.62	0.10	0.002	0.45	0.84
Garbage disposal (collection by sanitation service is ref.)					
Burned	0.92	0.16	0.640	0.65	1.30
Main source of drinking/cooking water (pipe is ref.)					
Well, pump (electric, hand)	1.35	0.21	0.055	0.99	1.84
Well water	1.69	0.33	0.008	1.15	2.49
Household size	0.93	0.03	0.028	0.87	0.99
Rural	0.32	0.06	0.000	0.23	0.45
Survey wave (wave 5 vs. 4)	0.44	0.05	0.000	0.35	0.55
Constant	0.43	0.10	0.000	0.27	0.68
Model 3 ^2^					
Agricultural household	0.50	0.09	0.000	0.36	0.71
Garbage disposal (collection by sanitation service is ref.)					
Burned	0.72	0.13	0.070	0.50	1.03
Main source of drinking/cooking water (pipe is ref.)					
Well, pump (electric, hand)	0.94	0.17	0.735	0.66	1.34
Well water	1.05	0.24	0.822	0.68	1.64
Household size	0.83	0.03	0.000	0.77	0.90
Rural	0.45	0.09	0.000	0.31	0.66
Survey wave (wave 5 vs. 4)	0.25	0.04	0.000	0.19	0.35
Constant	0.70	0.19	0.195	0.41	1.20
Model 4 ^2^					
Agricultural household	0.62	0.13	0.025	0.41	0.94
Garbage disposal (collection by sanitation service is ref.)					
Burned	0.77	0.16	0.203	0.52	1.15
Main source of drinking/cooking water (pipe is ref.)					
Well, pump (electric, hand)	1.07	0.22	0.727	0.72	1.59
Well water	1.20	0.32	0.506	0.71	2.02
Household size	0.79	0.04	0.000	0.71	0.87
Rural	0.42	0.09	0.000	0.27	0.64
Survey wave (wave 5 vs. 4)	0.35	0.06	0.000	0.25	0.49
Constant	0.41	0.12	0.003	0.23	0.75

^1^ The model includes a total of 1472 unique households (2816 observations over two cross-sectional survey waves in 2007 and 2014). ^2^ For variables not constrained to the proportional odds assumption, the model produces four sets of coefficients for the four cumulative logit models for comparing the odds of (1) not moved versus moved; (2) not moved or moved within the same village versus moved out of the village; (3) not moved or moved within the same district versus moved out of the district; and (4) not moved or moved within the same regency versus moved out of the regency.

## Data Availability

All data reported in this manuscript is publicly available from the RAND corporation website: https://www.rand.org/well-being/social-and-behavioralpolicy/data/FLS/IFLS.html (accessed on 27 August 2021). Additionally, the data can be made available upon request by contacting the corresponding author.

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
