# Peer review of "Household Flood Severity and Migration Extent in Central Java: Analysis of the Indonesian Family Life Survey"

_ijerph, 2023, doi:10.3390/ijerph20095706_

Round 1

Reviewer 1 Report

You have missed out some relevant citations regarding migration from:

OIM and IDMC

I suggest you included the most recent reports.

Author Response

Thank you for this comment. These are 2 very important organizations related to migration. We assumed OIM was a typo, and the reviewer was referring to IOM. We cited reports from both the IOM and IDMC in the introduction section (lines 37-43 and 49-52) and the discussion section (lines 292 and 362-365). However, we realized the references did not include the institution names. We have revised this. We have also explicitly stated the institutions where referenced in the introduction (lines 39-40 and 56-58).

We have also improved the description of the methods to make it clearer to readers. Please see the revisions on lines 99-141 and 194-217.

Reviewer 2 Report

This manuscript investigates how the severity of flooding influenced the extent of household movement in Central Java. They found that the more affected by the flood on the households, the less willing to move farther away. The topic of this paper is quite interesting.  But I am curious why the severely affected households are unwilling to move farther away. I think the authors could add the reasons to the abstract.

For the conclusions section, I think the authors should describe what they found from their study other than talk about what should be considered in further study.

Author Response

Thank you for raising this very important point. Unfortunately, we were unable to address the reasons why people did or did not move in this study because we did not have the data. We could only determine associations. While we did find that the most severely flood affected households tended to move less farther away, we do not know about their willingness or unwillingness to move. We agree that this is an important knowledge gap and we have added a line to the discussion section (lines 394-396). It would be interesting to have a qualitative perspective on why some households stay and others move. We suspect it’s because of the economic losses incurred from the disasters that make it more difficult to move for already vulnerable households, but this is speculation.

We agree that the conclusions should better communicate the study's findings. Please see revisions to the conclusions (lines 406-410).

Additionally, we’ve improved the methods and results sections to clarify the language. We have included a footnote to Table 2 to explain the 4 sets of coefficients given. Please see the revisions from lines 99-289.

Reviewer 3 Report

This exciting paper used a partial proportional odds model and analyzed the correlation between the severity of flooding and household movement in Central Java. The severity of the flooding was measured by the impact on households at the individual level, as defined by IFLS, which included death, injury, financial loss, or relocation of a household member. Households that suffered severe impacts from floods were found to have a 75% reduced likelihood of moving to a greater distance than those unaffected by floods. This suggests that the most severely affected households may remain in the flood-affected region. However, this paper will have more weight if it also includes data and analyses differences in migration between areas of the population living in rural and urban areas.

Author Response

Thank you. We are glad you enjoyed the paper and see its importance. Thank you for the comment. While we examined rural versus urban households in our model and found that rural households tended to move less farther away, we did not look at the differential effects in our original analysis. We agree this would be an important addition to the analysis. In response to this comment, we explored this by fitting a model with an interaction term to examine the differential effect of flood severity and rural/urban location on migration. However, due to very low cell counts, model convergence was not achieved. Therefore, we did not report this in our paper. We have, however, highlighted the finding about rural versus urban areas in the discussion and conclusions (lines 298 and 406-410).

Round 2

Reviewer 2 Report

The current version has been improved from the previous one.